# Undocumented: An examination of legal identity and education provision for children in Malaysia

**Tharani Loganathan**[1]*, **Zhie X. Chan**[1], **Fikri Hassan**[1], **Zhen Ling Ong**[2], **Hazreen Abdul Majid**[3,4]

**1** Centre for Epidemiology and Evidence-based Practice, Department of Social and Preventive Medicine, University of Malaya, Kuala Lumpur, Malaysia, **2** Department of Global Health and Development, London School of Hygiene and Tropical Medicine, London, United Kingdom, **3** Centre for Population Health, Department of Social and Preventive Medicine, University of Malaya, Kuala Lumpur, Malaysia, **4** Faculty of Public Health, Department of Nutrition, Universitas Airlangga, Surabaya, Indonesia

* drtharani@um.edu.my

**Data Availability Statement:** To protect the respondent's anonymity, ethical constraints prevent the data set from being made public. Data may contain personally identifiable or sensitive

## Abstract

Education is a fundamental human right. Yet there remain gaps in our understanding of undocumented children in Malaysia and their vulnerabilities in education access. This study aims to describe and contextualise undocumented children in Malaysia and their access to education. We conducted a desk review and in-depth interviews with 33 key stakeholders from June 2020 to March 2021. Framework analysis was conducted. Salient themes were geographical location and legal identity in terms of citizenship and migration status. We found that the lack of legal identity and non-recognition by the State was the root cause of vulnerability, experienced uniformly by undocumented populations in Malaysia. Only undocumented children with Malaysian parents or guardians can enter public schools under the Malaysian government's 'Zero Reject Policy'. Most undocumented and non-citizen children must rely on informal education provided by alternative or community learning centres that typically lack standardised curricula, resources, and accreditation for education progression beyond primary levels. Nevertheless, as non-citizen groups are diverse, certain groups experience more privilege, while others are more disadvantaged in terms of the quality of informal education and the highest level of education accessible. In Peninsular Malaysia, a very small proportion of refugees and asylum-seekers may additionally access tertiary education on scholarships. In Sabah, children of Indonesian migrant workers have access to learning centres with academic accreditation supported by employers in plantations and the Indonesian Consulate, whereas Filipino migrants who were initially recognised as refugees are now receiving little government or embassy support. Stateless Rohingya refugees in Peninsular Malaysia and Bajau Laut children at Sabah are arguably the most marginalised and have the poorest educational opportunities at basic literacy and numeracy levels, despite the latter receiving minimal governmental education support. Implementing a rights-based approach towards education would mean allowing all children equal opportunity to access and thrive in high-quality schools.

respondent information. Participants in the study are vulnerable populations whose data, when combined, could become identifying due to indirect identifiers (such as ethnicity, location, etc.). Data request could be made to the University of Malaya Research Ethics Committee (UMREC, reference number: UM. TNC2/UMREC- 848) at umrec@um.edu.my for researchers who meet the criteria for access to confidential data. Researchers can also contact the corresponding author for further details on the study methods and analysis. As the prescribed period by the Ethics Committee all information collected for this study will be kept safely for a minimum period of 5 years. Once the recommended period has lapsed without the need for any further analysis and audits, all electronic data will be deleted.

**Funding:** This research was funded by The Asia Pacific Observatory (APO) on Health Systems and Policies [grant number IF034-2020 awarded to Tharani Loganathan]. The funders had no role in study design, data collection and analysis, decision to publish, or preparation of the manuscript.

**Competing interests:** The authors have declared that no competing interests exist.

# Introduction

Education is a fundamental human right. The right of all children to free and compulsory education at the elementary and fundamental stages is affirmed in Article 26 of the Universal Declaration of Human Rights [1]. The Convention on the Rights of the Child (CRC) reiterates the right of all children to education on the basis of equal opportunity [2]. Malaysia has ratified the CRC. Yet, crucial reservations made to Article 28 paragraph 1 (a) have excluded non-citizen children from commitments towards universal education.

Malaysia has a robust public education system funded by the government with standardised national curriculums and examinations. Public education offers free or highly subsidised education to citizens from preschool, primary, secondary, post-secondary and tertiary levels [3]. There are also private options available mainly in urban areas for those who can afford the higher school fees [4–6]. Most undocumented and non-citizen children are not able to enter public schools [7] and must rely on informal education in the form of alternative or community learning centres supported by civil society organisations, faith-based organisations, private donors and local communities [8,9].

In Malaysia, the terms 'undocumented', 'non-citizen', 'migrant' and 'stateless' are sometimes used interchangeably. According to the Malaysian Immigration Act 1959/63 (Act 155), low-wage migrant workers are unable to bring their families or form new families in Malaysia [10,11], thus all children of migrant workers in Malaysia, irrespective of their parent's immigration status are irregular migrants. Also, Malaysia is not a signatory of the 1951 Convention relating to the Status of Refugees and its 1967 Protocol [12], nor has it ratified the 1954 Convention relating to the Status of Stateless Persons [13], or the 1961 Convention on the Reduction of Statelessness [14,15]. As a result, the immigration law does not distinguish between refugees, asylum-seekers, irregular migrants, undocumented or stateless persons, treating them as 'illegal immigrants'[11]. As such they are unable to gain legal employment and are at risk of detention and deportation, while access to healthcare, education, and social protection is severely restricted [16–18]. This is problematic as the current system risks criminalising those in need of international protection as well as penalising those who find themselves in these situations beyond their control.

Besides refugees and asylum-seekers who fall under the protection of the UNHCR in Malaysia [19], little is known about the other groups of undocumented children in Malaysia, often collectively referred to as the "invisible children" because of their exclusion from national databases. Thus, the scope and the complexity of educational exclusion in Malaysia may be underestimated [20–22].

According to the intersectionality theory, an individual's experience is shaped by the multitude of ways they are positioned according to race or ethnicity, class, gender, age, citizenship, language, religion, sexuality, and (dis)ability [23–26]. Thus, we hypothesise that although undocumented children are treated similarly by the state, they are part of diverse groups that confer different status, privileges, and marginalisation.

Education is an empowering right and an effective instrument for marginalised children to raise themselves out of poverty and fully participate in society [27–31]. Yet there remain gaps in our understanding of the diversity and commonalities within this hidden population of marginalised children and their vulnerabilities in education access. As such, this paper aims to define, describe, and contextualise the different groups of undocumented children in Malaysia and their access to education.

## Materials and methods

### Study design

This paper focuses on the undocumented and non-citizen children in Malaysia, including refugee and asylum-seeker, migrant, and stateless children. International students and children of expatriates were excluded from the analysis. We conducted a desk review of relevant literature and in-depth interviews with key stakeholders. Framework analysis was conducted to identify, define, and contextualise different categories of undocumented children at risk of education exclusion in Malaysia. Policy document reviews and qualitative analysis of stakeholder interviews allowed identification of issues and the implementation of policies. The findings presented in this paper were part of the broader study on non-citizen access to education in Malaysia.

### Data collection and analysis

We gathered academic journals, legal documents related to the right to education and citizenship, and policy documents relating to education provision for documents reviews. Reports from governmental agencies, international and local organisations were also gathered from official websites between June 2020 to June 2021.

For the qualitative component, data collection was conducted from June 2020 to March 2021. Semi-structured interview guides were initially developed based on our desk review. The interview guides contained introductory questions to understand participants' backgrounds and to contextualise non-citizen groups. Open questions were asked on education policies relevant to non-citizen children. These guides were developed for three main categories of interviewees: (a) teachers and educators, (b) parents and migrant representatives, and (c) policymakers and high-level stakeholders. We customised interviews according to the background of the interviewee. Minor revisions were made to the guide after initial reflections from the earlier interviews; see S1 File for interview guides. Participants were mainly recruited from LinkedIn, the professional networking website and through a snowball sampling technique. No further recruitment was made when the data does not yield additional insights. Participants were invited by telephone or email. Research information sheets and consent forms were sent before the interviews.

We purposefully interviewed stakeholders to represent the diversity of experiences of different populations throughout Malaysia. We conducted 32 in-depth interviews with 33 individuals. Most interviews were conducted on an individual basis; however, two interviews were conducted with 2 participants from the same organisation. One participant was interviewed twice.

Study participants were community organisers from civil society organisations, education providers from learning centres and schools, policymakers from government and international organisations, and academic researchers with professional expertise in the education of refugee, stateless, and migrant populations in Malaysia. We also interviewed adult refugees who had experience navigating the mostly informal education system in Malaysia as children. Our participants were able to share specific expertise on (1) stateless and undocumented children applicable to entire Malaysia, (2) refugees and asylum-seekers in Peninsular Malaysia, and (3) the migrant, stateless and undocumented communities in Sabah. Sample characteristics are shown in Table 1.

Due to public health measures in place during the COVID-19 pandemic, most interviews (28 of 33) were conducted remotely using an online platform (Microsoft Teams), while 5 interviews were conducted in-person in Kuala Lumpur, Malaysia. Most interviews were conducted

Table 1. Characteristics of the study participants (n = 33).

| Respondents' primary role | | Label | No. |
|---|---|---|---|
| Community organiser | | CO | 4 |
| Former students[1] | | FS | 7 |
| Education provider[2] | | EP | 11 |
| Policymaker | | POL | 4 |
| Researcher | | RES | 7 |
| **Total** | | | **33** |
| **Non-citizen type** | **Peninsular Malaysia** | **Sabah** | **Overall Malaysia** |
| Overall–non-citizens | | | 5 |
| Refugees | 15 | | |
| Stateless | 3 | 4 | 2 |
| Migrant[3] | | 3 | 1 |
| **Total** | **18** | **7** | **8** |

[1] All the former students interviewed were adult refugees. Of the 7 interviewed, 3 were also education providers.

[2] Of the 11 education providers interviewed, 7 also identified themselves as community organisers.

by at least 2 researchers, with one researcher leading and the other observing and taking field notes. Interviews averaged from 1 to 1.5 hours in length and were conducted either in English or Bahasa Malaysia (Malay language) depending on the participants' preference. All sessions were recorded and audios were transcribed verbatim. Transcripts were coded into emerging themes using NVivo 12 Plus (QSR International, Melbourne, Australia) and quotations were extracted into Microsoft® Excel® for Office 365 (Microsoft, Redmond, WA, USA). Interviews in Bahasa Malaysia were analysed in the same language, while quotes extracted were translated to English for publication.

We conducted a framework analysis; a qualitative methodology suited to applied policy research. Findings from the desk review and in-depth interviews were analysed using five steps: familiarisation, identifying a thematic framework, indexing, charting, and interpretation [32]. This descriptive analysis allowed us to categorise and contextualise undocumented children at risk of education deprivation by location (Overall Malaysia, Peninsular Malaysia, and Sabah, East Malaysia) and legal identities, define the concept of being 'undocumented', map the types of education provision and provide an analysis of key issues and policies on education provision that are specific or overlapping for each group of children. We triangulated our findings with desk research, incorporated perspectives of informed stakeholders from the interviews, and carried out bi-weekly discussions with the research team. Emerging themes from earlier interviews were probed further and validated in subsequent interviews. Results are presented with minimum quotations.

## Ethics

Informed consent was obtained before commencing interviews. Participants were informed that study participation was voluntary, and at any point would be free to refuse to answer questions or terminate interviews. All participants agreed to be audio recorded and quoted anonymously in publications. Audio recordings and electronic transcripts were stored in secure data servers.

This study was approved by the Medical Ethics Committee, University Malaya Medical Center (Approval numbers: UM.TNC 2/UMREC).

## Results

Study results are presented in two sections. In the first section, we describe categories of children at risk of educational exclusion. The second section describes education provisions for undocumented children in Malaysia. We describe different undocumented and non-citizen by their location in Malaysia based on our final framework (See Fig 1).

### Children at risk of educational exclusion

**Stateless and undocumented children in Malaysia.** The terms "stateless" and "undocumented" tend to be loosely defined and used interchangeably. For this paper, we use the definition of statelessness from the 1954 Convention relating to the Status of Stateless Persons as "*a person who is not considered as a national by any State under the operation of its law*" [13]. As such, the central concept of statelessness relates to whether a person has nationality or citizenship. Article 15 of the Universal Declaration of Human Rights states that "*Everyone has the right to nationality*"[1]. However, citizenship has been described as a privilege, and the attribution of citizenship by birth and rules for naturalisation varies by country. Malaysian citizenship laws are complex and certain children fall through the loopholes, thus are at-risk of statelessness.

On the other hand, being "undocumented" encapsulates broader scenarios that may apply to rightful citizens of the host country, non-citizens (foreign nationals), persons of undetermined nationality and those who are not citizens of any country (stateless), regardless of their place of birth (See Fig 2). In this study, we define an undocumented person as "*a person who*

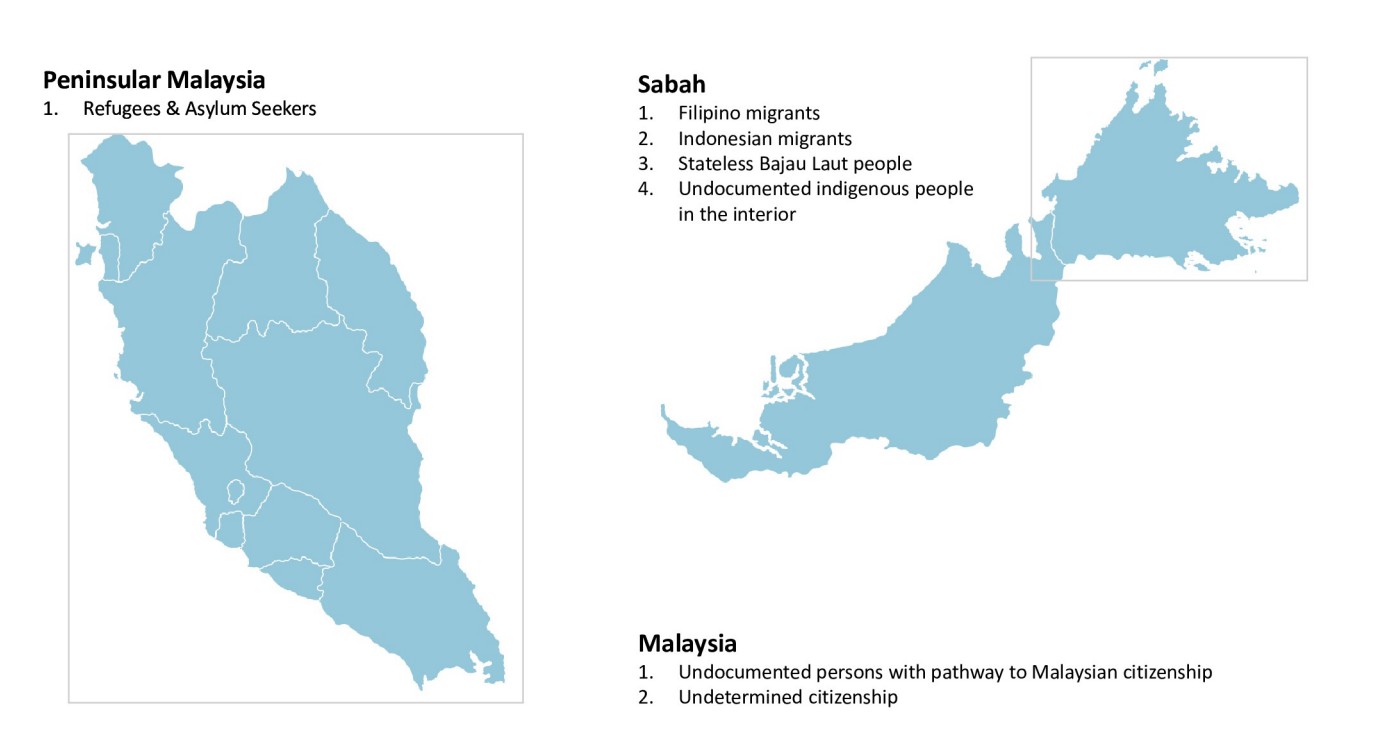

**Peninsular Malaysia**
1.   Refugees & Asylum Seekers

**Sabah**
1.   Filipino migrants
2.   Indonesian migrants
3.   Stateless Bajau Laut people
4.   Undocumented indigenous people in the interior

**Malaysia**
1.   Undocumented persons with pathway to Malaysian citizenship
2.   Undetermined citizenship

**Fig 1. Different categories of undocumented and non-citizens children in Malaysia by location and legal identities.** Note: Malaysia is comprised of Peninsular Malaysia and East Malaysia, separated by the South China Sea. Sabah is one of the states in East Malaysia. Reprinted from http://www.ofo.my/ under a CC by license, with permission from OFO Tech Sdn Bhd, original copyright 2021.

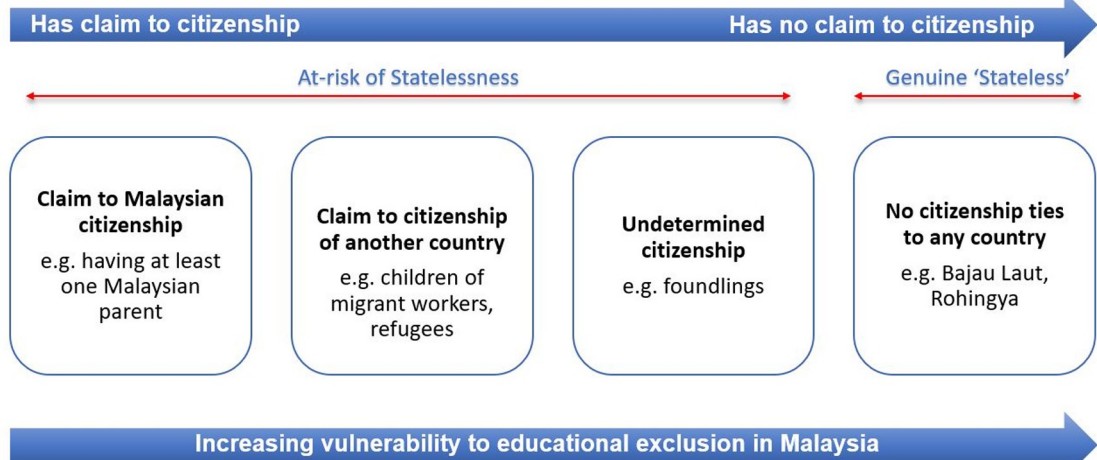

**Fig 2. Undocumented children in Malaysia according to risk of statelessness, claim to citizenship, and vulnerability to educational exclusion.**

*does not have the appropriate documentation to prove their legal status in the host country, restricting access to social services and movement.*" Undocumented persons may have incomplete or unrecognised identity documents or no documentation at all. It is acknowledged that the term "undocumented" is often used in the context of migration, referring to undocumented or irregular migrants [33]. However, our study found that people present in Malaysia not owing to international migration may also lack the appropriate documents to prove their legal status and face implications common to undocumented migrants. We have therefore expanded the definition of being "undocumented" to apply to non-migrants in this present study.

Fig 2 illustrates how undocumented children tend to be at greater risk of statelessness and educational exclusion depending on their claims to citizenship. We describe in the next section by the four categories of undocumented persons identified here: (1) Claim to Malaysian citizenship, (2) Claim to citizenship of another country, (3) Undetermined citizenship, (4) No citizenship ties with any country.

*(1) Claim to Malaysian citizenship.* For those born in Malaysia, factors contributing to the lack of legal status in Malaysia include the legitimacy of their parents' marriage, their mother's nationality, and the timeliness of birth registration.

Malaysian citizenship law is based on elements of the Jus Sanguinis ('right by blood') principle, whereby citizenship for a child is acquired through the citizenship of one or both parents. However, the situation is more complex as Malaysia does not give mothers and fathers equal rights to pass their nationality to their children [34,35].

In the specific case of Malaysian-born illegitimate children of a Malaysian father and a non-Malaysian mother (including traditional and customary marriages unrecognised by law), the child must follow the mother's citizenship. These children are at risk of statelessness, if the non-Malaysian mother is also undocumented or untraceable. The Malaysian father cannot confer his citizenship to illegitimate children, due to the current interpretation of citizenship laws which may be called gender-discriminatory [36,37].

Logistical difficulties and prohibitive travel costs from rural or interior regions may preclude the timely registration of births at Malaysian National Registration Departments (NRD) usually located in urban areas, placing some at risk of statelessness. Two groups affected are

the (i) indigenous and native people in the interior areas particularly in East Malaysia and the (ii) ethnic Indians in plantation estates, born or resident in Malaya since pre-independence, therefore eligible for Malaysian citizenship, but were unable to do so due to multigenerational historical marginalisation [38].

*(2) Claim to citizenship of other countries.* Migrants including regular or irregular economic migrants, refugees and asylum-seekers may register the births of their children born in Malaysia and obtain a birth certificate that classifies them as "non-citizens". However, birth registration does not confer citizenship entitlements, as Jus Soli ('citizenship by birth') is not practised in Malaysia. Migrant parents may neglect to register their child's birth as they are unaware that they are entitled to do so in Malaysia. More importantly, the fear of being detected by authorities may deter birth registration, as Malaysian immigration laws do not allow low-wage economic migrants to marry or have families in Malaysia. The lack of birth registration puts children at risk of statelessness, as a birth certificate provides proof of where a person was born and parentage–key information necessary to establish citizenship in any country. Children born in Malaysia would need to undergo citizenship verification at their embassies, to obtain citizenship of their home countries.

For non-Malaysian children, being undocumented goes beyond citizenship issues to encompass their legal right to stay in Malaysia. This also applies to refugee and asylum-seeking children who may already have citizenship from their home country, but do not have legal status in Malaysia as the state is non-signatory to the 1951 Refugee Convention and its 1967 Protocol [12].

*(3) Undetermined citizenship.* Persons with undetermined citizenship includes foundlings or children abandoned at birth with no documents. By law, Article 19B of the Second Schedule of the Federal Constitution states that "*any newborn child found exposed in any place shall be presumed, until contrary as shown, to have been born in that place to a mother permanently resident there*".

In practice, foundlings are given temporary residence status (MyKas) that is renewable every five years, but only if they are registered by the Department of Social Welfare. Furthermore, Article 14(1)(b), Section 2(3) of the Federal Constitution also states that a "*child born in Malaysia who is not born the citizen of another country and who cannot acquire citizenship of another country within 12 months is a Malaysian citizen*" [39].

However, this is not being implemented for foundlings. Foundlings have difficulty with birth registration as this requires two witnesses to prove their parent's identity and place of birth. In these circumstances, registration of birth would need approval from the Minister of Home Affairs and could take up to five years, further complicating citizenship application.

In cases of adoption, adoptive parents are unable to pass their citizenship on to adopted children. Both the Registration of Adoptions Act 1952 (Muslim) and the Adoption Act 1952 (Non-Muslims) are silent on the question of citizenship of adopted children.

*(4) No citizenship ties with any country.* Only children without claim to citizenship of any country are considered truly "stateless". The stateless groups in Malaysia are the Bajau Laut, a seminomadic seafaring tribe from Sabah, East Malaysia, and the Rohingya people who are also refugees fleeing persecution from Myanmar. Both groups have faced generations of marginalisation and stigma.

**Refugees and asylum-seekers in Peninsular Malaysia.** Refugees are persons forced to flee their country of origin because of persecution, war or violence and are unable or unwilling to return due to a well-founded fear of persecution [40]. Asylum-seekers are individuals seeking international protection, but whose refugee status has yet to be determined [41].

As of August 2021, 179,390 refugees and asylum-seekers were registered with UNHCR in Malaysia [42]. The majority were Rohingya (57%), Chin (13%), and other ethnic groups (16%)

from Myanmar (86%). The remaining 14% of refugees and asylum-seekers come from 50 other countries, including Pakistan, Yemen, Syria, Somalia, Afghanistan, Sri Lanka, Iraq, Palestine, and others. Refugees and asylum-seekers reside in urban, non-camp-settings in Peninsular Malaysia, with most located within the Klang Valley (Selangor and Kuala Lumpur) (56%). According to the UNHCR, of the 23,823 persons of concern of school-going ages in 2017, only 30% are enrolled in community learning centres [43].

As Malaysia is not a signatory to the Refugee Convention 1951 and its 1967 Protocol [12] the status of refugees is not recognised by the state and the state holds no legal obligation to protect them. Under Malaysia's Immigration Act (Act 155) [11], refugees and asylum seekers are deemed "illegal migrants" and are subject to arrest and detention.

Refugees and asylum-seekers travel to Malaysia via air routes (with passports), or through arduous land or sea routes (without passports). Following arrival, they would undergo UNCHR Registration and then the Refugee Status Determination (RSD) which is a lengthy, rigorous process to confirm refugee status [44]. The UNHCR identity card issued when refugee status is confirmed offers limited protection against the risk of arrest, while allowing for discounted fees at public healthcare [45,46], access to education and other essential services provided by UNHCR and partners. Unfortunately, the UNHCR identity cards have no formal legal value in Malaysia [44], thus, refugees and asylum-seekers may still be detained in immigration detention centres.

In general, refugees and asylum-seekers have three long-term solutions: (i) voluntary repatriation to their home country, (ii) local integration in the host country, and (iii) resettlement to a third country [19]. As a non-signatory to the 1951 Refugee Convention, local integration is not a formal option for refugees in Malaysia. Malaysia is therefore considered a transit country for refugees and asylum seekers. However, as repatriation is only possible for some and resettlements are rare, refugees tend to experience protracted stays in Malaysia. This leaves refugees in uncertainty, potentially for decades, without formal, robust protection or the provision of basic human rights, such as the right to work, security, healthcare, and education.

*Identity, marginalisation, and privilege.* The most vulnerable among the refugee population is arguably the Rohingyas, who not only had to flee from systematic socio-historical persecution but are also stateless. As such, they are not recognised as citizens of any country and have nowhere to return to. First arriving in Malaysia in the 1990s, the Rohingyas are the largest refugee group in Malaysia with over 140,000 people [47]. Compared to other refugee groups, the Rohingyas have been in Malaysia for several generations and over three decades among Malay-speaking communities. They are typically fluent in the spoken Malay language, which helps them to integrate with the local population to a certain extent. There have also been cases of "nominal adoption", where Malaysian locals adopt Rohingya children to enable some access to public education while allowing the children to remain with their biological family. Rohingya children may access refugee learning centres, madrasahs or tahfiz run by the Rohingya communities. Nevertheless, interviewees highlighted that the Rohingya children have poorer educational attainment as they experience more prolonged marginalisation and discrimination compared to other refugee groups. This, along with the practice of child marriage, may influence the lack of parental support for post-primary education, especially for girls.

In contrast, Middle Eastern refugees tend to hold more positive attitudes towards education compared to Rohingya refugees. Middle Eastern refugees tend to be of a higher socioeconomic status, more educated and may have held professional jobs in countries of origin. Consequently, there are different levels of school readiness, educational experiences, family support and expectations for children of the same age group.

Study participants also informed that different refugee communities tend to be treated differently in Malaysia and this might contribute to their experiences in seeking education and accessing services in Malaysia.

*'In Malaysia, which group of refugees you are [part of] also determines how you are being treated, right? So, again, this is whole layers and levels. Even [amongst] refugees, you will find, like say the Bosnians or Syrians, may be treated more favourably. And this is not the ideal at all. Even when we say more favourably, it doesn't mean that they can work here. In terms of the stigma—there's less stigma for these groups than say for Rohingya for example.'* POL-02

**Undocumented children in Sabah.**   The state of Sabah in East Malaysia is unique due to porous national borders, intergenerational cross-border migration with the neighbouring Philippines and Indonesia, and ongoing disputes with the Philippines government regarding territorial claims in Sabah [48,49]. The Department of Statistics Malaysia estimated in 2021 more than a quarter of the population of Sabah are non-citizens (995,400), of which about 250,000 are children aged from 0 to 19 years [50].

The subgroups at risk of educational exclusion include the children of (1) poorly documented Filipino migrants, (2) regular and irregular economic migrants from Indonesia, (3) stateless, semi-nomadic Bajau Laut tribes and (4) undocumented local people in interior regions who face geographical barriers to birth registration. While these groups have different levels of marginalisation and privilege because of their identity and entitlements, the long history of mixed marriages between the migrant groups from the Philippines and Indonesia and local communities have blurred boundaries, making the migration situation complex.

Interviewees suggested that issues concerning children's educational exclusion were less dire in the neighbouring state of Sarawak, where education exclusion was exclusively among undocumented communities in the interior bordering Kalimantan, Indonesia.

*(1) Poorly documented Filipino migrants.* Sabah hosts many poorly documented migrants whose ancestry can be traced back to the Southern Philippines. Three migration waves from the Southern Philippines are described: (i) the first wave was the traditional and historic free movement of the coastal peoples that occurred since before the formation of Malaysia in 1963, (ii) the second wave was driven by the conflict in Southern Philippines in the 1970s, and (iii) the third wave was associated with economic migration in response to Sabah's post-1990s economic development policies, alongside labour migration from Indonesia [51].

They have since settled in Malaysia for several generations, many of whom were born and raised in Sabah and no longer have contemporary social ties in the Philippines. Multiple generations of intermarriages and poor documentation obscure the pathways to citizenship and increase their risk of statelessness [52]. In 2016, it was estimated that there were 136,055 Filipino migrants in Sabah, and around 50,000 held IMM13 visas issued in the 1980s [53].

We found that the local Sabah people do not refer to the Filipino community as "refugees" but as the "undocumented" or "stateless" persons in Sabah. They are no longer recognised as refugees under UNCHR's operation in Malaysia [54]. And yet, initially, those fleeing conflicts were given an IMM13 refugee visa, a special document produced under Form 16 of the Malaysian Immigration Act [11] that confers temporary rights to stay, work and access public education [55]. However, these rights have gradually been eroded.

The IMM13 visas were issued in the 1980s to the male head of the family to include the dependent wife and children. Subsequent generations did not receive the IMM13 visa. Also, many allowed the visas to lapse, as they were unaware that IMM13 visas needed annual renewal or were unable to pay the renewal fees. In addition, the issuance of two other identity documents that do not confer citizenship benefits (*Kad Burung-burung* issued in the 1980s

and 1990s and *Kad Banci* or *Kad Hijau* issued in the 2000s) and the rampant use of fraudulent identity documents served to confuse the documentation issue [52,56].

Recently, plans to consolidate the three identity documents under a "Sabah Temporary Pass" had been dropped following opposition from the local community [57,58]. Due to the alleged "Project IC", a citizenship-for-votes scandal [59], the local Sabahan community tend to be suspicious with initiatives to provide identity documents or perceived citizenship entitlements like healthcare or education access to non-citizens.

Our interviewees suggest that those of Filipino origin are among the most disadvantaged of the different subgroups in Sabah. Compared to Indonesian labour migrants, those of Filipino origin have been in Sabah for longer, are greater in numbers and tend to settle in urban areas rather than in oil palm estates. This stands in contrast with Indonesian labour migrants and their families, whose welfare is formally cared for by estate managers and have well-tended living conditions, with greater involvement of the Indonesian embassy. Those with Filipino origins are also more at risk of detention and deportation, which sometimes involve parent-child separation. In addition, the Philippine Consulate only has offices in Kuala Lumpur, but not in Sabah due to the frayed geopolitical situation, further exacerbating their risk of statelessness.

*(2) Indonesian labour migrants in the oil palm plantations and urban areas.* Economic migration from Indonesia to Malaysia is not restricted to Sabah, but also in Sarawak and Peninsular Malaysia [60,61]. Under Malaysian Immigration Laws, low-wage economic migrants—commonly employed via the Employment Pass Category III (monthly salary below RM5000 and Temporary Employment Pass)—are not allowed to bring dependents, nor are they allowed to marry and establish new family units in Malaysia [62]. East Malaysia is the exception, as it is understood that migrant workers employed in estates (the majority of whom are Indonesian males) may bring their spouse and children if the estate can ensure the children's education and the prevention of child labour in compliance with Roundtable Sustainable Palm Oil (RSPO) certification requirements [9,63].

It was estimated that there are between 30,000 to 60,000 children of Indonesian migrant workers in Sabah [9,64]. Interviewees suspect that there may be similar numbers of Indonesian children in Peninsular Malaysia, but we were unable to verify these figures.

In Peninsular Malaysia, migrant workers are concentrated in urban areas, while in Sarawak, most reside in plantations. In Sabah, our study participants informed that half the Indonesian population may be found in plantations, while the other half are in urban areas, working at small farms or businesses. Mid- to large-sized oil palm estates collaborate with the Indonesian embassy to build and maintain learning centres within their estates. Otherwise, estates must provide children with transportation to the closest learning centres. Besides educational provision, it is also noteworthy that estates also provide housing, access to healthcare and amenities for migrant workers and their families. Study participants also informed that Indonesian children have less issue in documentation because of the active presence of the Indonesian Embassy in Sabah. NGOs with the support of employers, work with the Indonesian embassy to facilitate the issuance of birth certificates.

*(3) Stateless Bajau Laut nomadic tribes.* The Bajau Laut—also known as Sama Dilaut, Pala'o or Sea Gypsies—reside in Sabah's East Coast. The Bajau Laut are a seafaring community, who historically roamed between the islands off Sabah and the Sulu archipelago, that is part of present-day Philippines. Although they have been living along the coast of Sabah for generations, this mobile community are genuinely "stateless", as they are not regarded as citizens of any country [65]. The Department of Statistics Malaysia estimates that there are 468,000 people from the Bajau ethnic group in Sabah. However, estimating the Bajau Laut population is challenging due to their semi-nomadic lifestyles [66].

The Bajau Laut people are largely unable to speak the national language (Bahasa Malaysia) and instead use their own distinct language (Bajau or Samal) and hold firmly to the traditional customs, rituals, and beliefs. Due to increasing security and environmental restrictions, the Bajau Laut people are increasingly forced to abandon their traditional, semi-nomadic lifestyle and marine-based income. While traditionally living in houseboats, many now live on wooden stilt houses built over the sea, without access to clean water and electricity. Being stateless and without legal identity documents, the Bajau Laut people are excluded from state-provided education and healthcare services. Particularly marginalised, they are stuck in an intergenerational cycle of poverty, are excluded from formal employment, and are frequently engaged in begging and glue-sniffing [56,67,68].

*(4) Undocumented indigenous people in the interior.* The indigenous peoples of Malaysia (collectively known as the Orang Asal), were estimated at around 13.7% of the country's population in 2018. Diverse indigenous ethnic groups form a substantial proportion of the population in Sabah and Sarawak. Sarawak's indigenous population (the Dayak or Orang Ulu) constitutes 71% (2 million) of Sarawak's population, while the indigenous people (natives or Anak Negeri) make up 59% (2.3 million) of the Sabah's population. In contrast, the indigenous peoples of Peninsular Malaysia (Orang Asli) are a minority group of 0.7%(198,000) of the population [69,70].

The indigenous people who generally live in the remote, interior regions are at risk of statelessness due to untimely registration of births. Factors that contribute to the risk of statelessness, include (1) the financial costs and geographical difficulties traveling to the National Registration Department usually located in towns, (2) the lack of awareness of the parents on registering births, (3) traditional and customary marriages not recognised by the law, and (4) home births. The border communities face additional challenges associated with cross border migration and marriage with non-citizens, where children born only have citizenship if the mother is a citizen.

To help resolve documentation issues in the interior regions, the Sabah state government runs a mobile court, where village heads serve as witnesses for citizenship applications by confirming that a family has stayed in their village for several generations. Interviewees commented that resolving documentation issues among locals from the interior is more straightforward, compared to the undocumented people of non-citizen origin or the Bajau Laut community.

## Education provision for undocumented children

Prior to 1990s, non-citizen children were able to enter public schools on a case-to-case basis through the discretion of the school heads. Subsequent policy changes in 1990s, imposed annual fees for non-citizen children and more stringent identity document requirements for school entry, making it difficult for undocumented and non-citizen children to enter public schools.

In Malaysia, education provision is governed by the Education Act 1996 (Act 550). The 2002 amendment of the Education Act made 6 years of primary education compulsory for all Malaysian citizens aged from 6 to 12 years, without mention of non-citizen children [71].

Nevertheless, the government has permitted the establishment of an informal or alternative education system, through learning centres run by NGOs and other stakeholders.

The National Education Policy of 2017 states that primary education is compulsory for all children aged 6 to 12 years, and this includes non-citizens. Also, the 2018 'Zero Reject Policy' allows poorly documented children of Malaysian parents entry into public schools on the condition that the children have initiated citizenship applications. See Fig 3 for the timeline of the

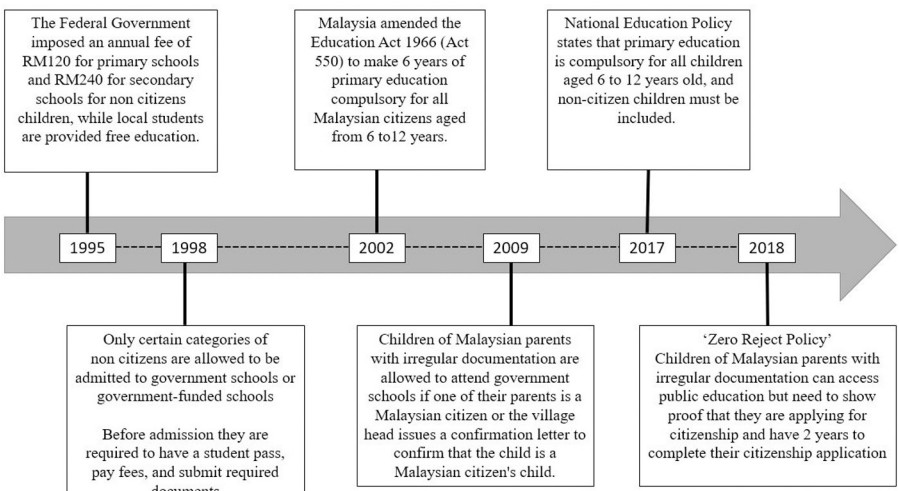

**Fig 3. Timeline of the evolution of education policies for non-citizens children in Malaysia. Sourced from [71–76].**

evolution of education policies for non-citizen children in Malaysia and Table 2 for the types of education access available for these children.

**Stateless and undocumented children in Malaysia.** We found that children with a claim to Malaysian citizenship have more educational opportunities compared to other undocumented children with minimal access to public education (See Fig 2). Children with a claim to Malaysian citizenship but with irregularities in documentation have limited entitlements to public education through the Zero Reject Policy and Sekolah Bimbingan Jalinan Kasih.

*Zero Reject Policy.* The Zero Reject Policy was initiated by the Ministry of Education in 2019, to facilitate undocumented children's entry into public schools [73]. This policy is in line with the aims of the national education policy to ensure that all children have access to education. Nevertheless, only certain undocumented children can gain entry to public schools. The eligibility criteria for public school entry are as follows: 1) non-citizens adopted by Malaysian citizens; 2) illegitimate children of a Malaysian father and a non-Malaysian mother; 3) non-citizen children whose parents who are also non-citizens (parents are foreign embassy staff, parents working at a government agency with a valid work permit, parents are permanent residents in Malaysia, foreign children sent to continue education in Malaysia as part of a government exchange).

According to the Zero Reject Policy, undocumented children are allowed to enrol in public schools with proof of guardianship (at least one Malaysian parent) and the understanding that parents or guardians would submit all identity documents within 2-years from the start of schooling (in the process of applying for Malaysian citizenship).

According to our interviewees, the Zero Reject Policy has a limited reach, as it is only applicable to a small subset of undocumented children with a claim towards Malaysian citizenship.

*"Technically, efforts have been taken to ensure that if you are a native population and the issue is merely documentation issues—You are a citizen, it's just that you don't have the documentation to prove it maybe. The treatment is not the same if you are a migrant worker's child without documentation–the treatment is different. Yeah, in that circumstance the chances of going to government schools are just very remote—It is not possible."* RES-04

**Table 2. Types of education access for undocumented and non-citizen children in Malaysia.**

| Categories of non-citizens children in Malaysia | Types of Education | | | | | | | |
| --- | --- | --- | --- | --- | --- | --- | --- | --- |
| | Formal Education | | | Informal Education | | | | |
| | Public School | Private School | | Learning Centre[1] | | | | |
| | | International school[2] | Expatriate school[3] | Malaysian Government-supported[4,5] | Embassy supported | | NGO or Community operated[8] | Islamic religious school (Madrasah or Tahfiz)[9] |
| | | | | | Indonesian[6] | Philippines[7] | | |
| **Overall Malaysia** | | | | | | | | |
| Stateless and undocumented with one Malaysian as parent or guardian | √ | | | √ | | | √ | √ |
| **Peninsular Malaysia** | | | | | | | | |
| Refugee and asylum seekers | | √ | | | | | √ | √ |
| **Subcategories in Sabah** | | | | | | | | |
| Filipinos | | | | | | √ | √ | √ |
| Indonesian | | | √ | | √ | | √ | √ |
| Bajau Laut | | | | √ | √ | | √ | √ |
| Local people in the interior[10] | √ | | | | | | √ | |

[1] Learning centres are an alternative pathway for children lacking the necessary identity documentation to gain access to formal education at public schools.

[2] Private international schools are only accessible to those with the financial means to pay the school fees. Valid passports and visas are required for admission to international schools.

[3] Expatriate schools are formal schools established under the auspices of an embassy. Examples of expatriate schools are the Sekolah Indonesia Kuala Lumpur, Johor Bharu and Kota Kinabalu, which informally accept children of migrant workers.

[4] Sekolah Bimbingan Jalinan Kasih (SBJK) is an alternative education programme under the Ministry of Education, Malaysia that caters to abandoned and street children in Malaysia.

[5] The National Security Council (NSC) of Malaysia, in collaboration with UNICEF and other organisations, provides alternative education for undocumented children in Sabah. As of June 2015, there were 12 NSC run learning centres in operation.

[6] Malaysia and Indonesia signed a Government to Government (G2G) Agreement in 2006 to ensure that children of Indonesian migrant workers have access to education. The Indonesian government sends in qualified Indonesian teachers to teach in learning centres in Sabah and Sarawak.

[7] In 2014, six learning centres in Sabah signed a Memorandum of Understanding on a community-based education program with the Commission on Filipinos Overseas (CFO), the Department of Education (DepEd), and the Philippine Embassy in Kuala Lumpur.

[8] Refugee or asylum-seekers access informal education through learning centres, either partnered with UNHCR or entirely community-based. As of March 2020, there are 126 learning centres recognised by UNHCR in Peninsular Malaysia.

[9] Islamic religious schools in Malaysia, also known as *Maahad Tahfiz*.

[10] Many indigenous children are unable to attend school because of distance, lack of documentation, or language barriers.

Sourced from [3,21].

Our interviewees based in Sabah concurred that the policy has less relevance in their context, where many undocumented children do not meet the eligibility criteria linked to parents' Malaysian citizenship.

*Sekolah Bimbingan Jalinan Kasih*. For homeless or abandoned street children with incomplete documents and some claim to Malaysian citizenship (at least one Malaysian parent), the Ministry of Education opened a dedicated public school, Sekolah Bimbingan Jalinan Kasih (SBJK) located in Chow Kit, Kuala Lumpur, with plans to expand this school model to several other states [77]. The SBJK serves children aged 5 to 17 years old, who are mainly street children living near its location in central Kuala Lumpur, marginalised due to poverty and other social problems and who are unable to enter mainstream public schools due to the lack of identity documents.

Interviewees informed that while the school aims to give undocumented children a basic education to improve their job prospects, their lack of documentation does not allow them to pursue tertiary education or formal employment.

*"But the problem is like this—In our school, even though the student is good academically, but he has no documents, therefore even if he wants to enter a IPT [Institute of Higher Learning or a public university], he cannot because he is undocumented or has incomplete documents. Because of that it is even difficult to get a [decent] job because there are no documents. That's the problem!"* POL-04 Translated from the Malay Language

*Undocumented children of undetermined citizenship.* Children abandoned at birth or 'foundlings' are unable to prove their parentage. Thus, they are of undetermined citizenship and are particularly vulnerable to statelessness. Our interviewees informed us that even 'foundling' children residing in government welfare homes were unable to register in public schools, due to incomplete documentation. Instead, some undertake vocational training within the welfare institutions, although they are unable to receive the Sijil Kemahiran Malaysia (Malaysian Skills Certificate), unlike their counterparts with Malaysian citizenship.

Interviewees also informed us that despite the efforts of the government welfare institutes, some undocumented children who were institutionalised from birth may leave care at the age of 18 years, without obtaining Malaysian citizenship.

**Refugees and asylum-seekers in Peninsular Malaysia.** Refugee and asylum-seekers access informal education through learning centres, either partnered with the UNHCR or entirely community-based. Other options include Madrasah or Tahfiz schools that focus on Islamic religious education (run separately from the religious schools catering to Malaysians) and private international schools for those who could afford them. Although some have accessed public schools in the past, this is not an option for refugee and asylum-seeking children at present.

*Learning centres for refugees.* Learning centres are not regulated by the government, nor do they have a standardised syllabus, teachers qualifications requirements, fee structures or admission policies resulting in a great variety in quality. Learning centres catering for refugees may be independent or registered with the UNHCR. As of March 2020, there were 126 learning centres recognised by UNHCR in Peninsular Malaysia, with most located within the Klang Valley (84%). Nearly all learning centres provide pre-primary (86%) and primary education (94%), but only one-fifth (21%) provide secondary education. While the majority use the English language as the medium of instruction (95%), many centres also teach Bahasa Malaysia or refugee languages [78].

The UNHCR advocates for refugee children's access to education by providing financial and material support to recognised learning centres. UNHCR acts as a coordinator to enhance the quality of education for refugees by building capacity for teachers [43]. All learning centres recognised by UNHCR are issued protection letters, stating that the agency is fully aware and supports initiatives to provide education for the refugee community. The letter also states that students attending the centres are 'persons of concern', which allows students some protection from harassment. The UNHCR has identified six implementing partners to serve its objectives in improving access and learning opportunities for the refugees living in Malaysia. These key partners tend to be larger education providers with stronger track records compared to smaller learning centres. A key partner may have enrolments of up to 1,800 students at a given time.

Learning centres are not allowed to use the Malaysian syllabus, as they are not recognised as schools by the Ministry of Education. We found that some learning centres may use a curriculum that is best described as *'mix and match'* by interviewees. The syllabus used may be based on Malaysian or other countries' national syllabus and is influenced by the availability of textbooks and other resources, and instructors' familiarity with the material. Education providers informed that learning centres cater to the needs of the community, and for some refugee

communities it would suffice to provide basic language for communication with locals and informal employment.

And yet, interviewees revealed that refugee education often emphasises preparation for resettlement in English-speaking countries such as the USA, Canada, Australia, and others. In practice, this means learning centres prioritise the use of the English language and international syllabi (IGCSE or British O levels) wherever possible. Unfortunately, the limiting factor is the prohibitive fees for international examinations, and children must rely on private sponsors for financial support. Therefore, only a handful of students, at select learning centres sit for IGSCE examinations.

Refugees and asylum-seekers tend to attend schools run by their communities. Their differing backgrounds confer different levels of social or cultural capital to education access. Interviewees explained that although learning centres using international syllabi are sought after by non-Rohingya communities, all refugees should aspire for better education for resettlement.

> *"But the non-Rohingya [refugees], the Arabs, the Africans, they sit for international syllabus, and they need to. Actually, even the Rohingya should. Because ultimately, you're going to be resettled, right? So, whether they sit for the local syllabus or the international syllabus, at the end of the day, what they can do with it is limited, right? What can they do with the certificate?"* CO-01

Non-citizen children are unable to sit for Malaysian school leaving certificate examinations and learning centres 'certificates of completion' are not recognised in Malaysia or elsewhere. Hence, this is the inevitable "glass ceiling" that hinders entry into tertiary education and better career prospects for non-citizen children in Malaysia. Together with the lack of the right to work in Malaysia, the lack of recognised education is demotivating for children.

The inability of refugee parents to obtain formal employment also indirectly affects children's access to education. We found that it is relatively common for refugee and asylum-seeking students to drop out of school at the post-primary level, to help supplement the family income. This may also explain the paucity of refugee learning centres that offer secondary education in Malaysia.

*Islamic religious schools*. Islamic religious schools (madrasah or tahfiz) in Malaysia may be broadly classified as state-funded or private religious schools. State-funded religious schools are financed either by the federal government (under the Ministry of Education) or by state governments (registered with state religious authorities). These schools are for Malaysian citizens and follow the national curriculum. Private religious schools receive private funding and are not under the jurisdiction of the Ministry of Education [79,80]. Some community-run private religious schools cater solely to refugees, and several of these are registered with the UNHCR and receive partial oversight from religious institutions such as JAKIM (Department of Islamic Development Malaysia).

According to interviewees, many religious schools catering to the Rohingya community are unregistered and unregulated. Religious schools attended by Rohingya may be operated by community members who were religious teachers back in Myanmar, but there is no standardisation of teachers' credentials.

Those interviewed expressed concern about the quality of education at religious schools as instruction concentrates on Quranic recitation and religious education and may not include a standardised academic curriculum. However, enrolment in a religious school may be the only viable option for refugee parents especially if they are unaware of alternatives. These religious schools are responsive to the Rohingya community's need for religious instruction in their native language and are affordable.

*"I feel like parents are sending children to religious education because they can't afford the children to be at home. In religious school, the Tahfiz school is feeding the children. So, they have this mindset of thinking, 'Okay! My child can't go to normal school because I can't afford it and it is not accessible, whether it's far, or it is another ethnic community's school or something. But at least they can have religious education. So, when I die, he will read Quran for me.' Even if you need to pay, it is only a one-time payment per year."* FS-01

*Refugees' access to public education.* We found that there have been instances in the past where refugee children had entered public schools. Policy changes since the 1990s (See Fig 3) have meant that more stringent entry requirements, together with the recent advances in digitalisation of school registrations, have made it near impossible for refugees and asylum-seekers to enter public schools.

*"In the past, the headmaster had discretionary power. But because now everything is online, if the document is not complete, the system would automatically reject. Now the Headmaster does not have any power. He only accepts what is given by the District PPD [District Education Office]. If those children have problems [with school registration], they must go to the PPD, or even to the JPN [State Education Department] to register and to apply. It is very rigid."* RES-07 (translated from Bahasa Malaysia)

We interviewed several refugees who experienced education at public primary schools. Non-citizen children pay higher school fees, are not eligible for the free textbooks program and are unable to sit for school-leaving examinations. Yet, students may otherwise participate in lessons with Malaysian children, gain local language skills and experience a more structured, higher quality education compared to that offered at community-run learning centres. However, experiences at public schools were mixed. Some interviewees shared experiences of bullying and being unable to fit in at public schools.

**Undocumented children in Sabah.**   Alternative education in the form of learning centres has emerged to serve the educational needs of the disadvantaged children in Sabah. These learning centres are supported by NGOs, faith-based organisations, local communities, government agencies (National Security Council, the Malaysian Armed Forces, and others), employers' groups (oil palm plantations), and foreign embassies (Indonesian or Philippines embassies).

Overall, learning centres in Sabah may be classified as either located within or outside of palm oil plantation estates. The most organised and vigorous initiatives for alternative education appeared to be for the children of Indonesian migrant workers, largely served through the learning centres in estates. Learning centres outside estates are primarily managed by NGOs or local communities in urban areas and have fewer resources. Education providers informed that urban learning centres face harassment and repeated closure by the state government, as they can't meet the Ministry of Education's (MOE) criteria for schools.

*"We have enormous problems with the local education department. The reason is that they regard all learning centres, and there are now several 100 learning centres that are independent of the [Indonesian] consulate, that are just citizens setting up learning centres. . . They regard them as illegal, as they do not adhere to the law that is set out by MOE. Okay, I have been summoned to the Education Department for this purpose too! And I have had schools shut down by the local education authorities! Last year, I had one school being shut down three times."* EP-04

Unlike the refugee learning centres in Peninsular Malaysia, most learning centres in Sabah teach almost exclusively at pre-primary and primary levels and have modest expectations of providing basic education, vocational skills and preventing social ills.

*Embassy supported education for children of migrant workers.* The educational needs of children of migrants are partly catered for by the government of their respective countries of origin, notably via the Indonesian embassy (through a Government-to-Government agreement) [81] and to a lesser extent the Philippine embassy (through a Memorandum of Understanding with several Sabah-based learning centres) [82].

Compared to children of Filipino descent, there is additional support for Indonesian children from oil palm estates in Sabah. Indonesian law stresses the importance of educating their citizens wherever they may be located. Article 6 (1) Law of the Republic of Indonesia No. 20 of the year 2003 on the National Education System states that *"Every citizen aged seven to fifteen years, wherever they are, both within the country and outside the Republic of Indonesia, shall attend basic education."* On this basis, a collaboration between the Indonesian government, oil palm plantation companies, and local NGOs began in the mid-2000s, to establish learning centres for the children of migrant workers. The Indonesian government would provide trained teachers and pay their salaries, while oil palm estates would contribute towards school facilities and teachers' housing. These learning centres follow an existing, robust alternative curriculum used in Indonesia for older children and dropouts from mainstream education.

An important implication of the collaboration with the Indonesian government is that these children are given the option for further education after leaving the learning centres. Since the Indonesian Ministry of Education recognises the certificates issued by recognised learning centres, students may be repatriated to Indonesia to pursue secondary or even tertiary education on government scholarships, which would lead to improved job prospects.

According to our interviewees, the learning centres within estates are better resourced in terms of funding, infrastructure, and teachers compared to learning centres located outside estates. Even with the Embassy support, alternative education for Indonesian migrant workers' children is less accessible outside of estates due to high rental costs and lack of assistance from urban employers.

Interviews with the Indonesian consulate revealed that there were over 300 learning centres in Sabah and Sarawak catering for an estimated 30,000 Indonesian children of school age. Learning centres are not allowed to formally operate for children of Indonesian migrant workers in Peninsular Malaysia, highlighting a huge unmet need for education. To date, three informal learning centres (*Sangar Bimbingan*) with a more basic syllabus operate in Peninsular Malaysia under the auspices of the Indonesian consulate. Besides these, there are also three Indonesian international schools (*Sekolah Indonesia*) located in major cities in Sabah and Peninsular Malaysia, which cater primarily to Indonesian expatriates' children, although children of Indonesian migrant workers have limited access to these schools.

The Philippines government plays a more modest role in supporting their nationals in Sabah. In 2014, six learning centres in Sabah signed a Memorandum of Understanding with the Commission on Filipinos Overseas (CFO), the Department of Education (DepEd), and the Philippine Embassy in Kuala Lumpur to collaborate on a community-based education program in Sabah. The Philippine Embassy offers assistance to these learning centres in terms of teacher training, resources, and fundraising efforts [82]. Yet, these learning centres are not recognised by the Malaysian government and are only allowed to operate on the condition that sessions are limited to a 3-day week and school uniforms are not allowed.

*Learning centres for the Bajau Laut nomadic tribes.* The 2002 amendment of the Education Act had excluded all undocumented children, including children from the Bajau Laut, from entering public schools in Sabah. Interviewees informed that it was challenging to provide

education to this community, as providers would need to cater to their semi-nomadic lifestyle and lack of familiarity with the national language. Nevertheless, several local NGOs attend to the community's basic educational needs by operating mobile teaching units that service the settlements [21]. One local NGO even converted a floating Bajau house into a learning centre. This interviewee stated that obtaining a safe space for a school is a major challenge faced by NGOs in helping the community.

> *"Most of these houses on the water don't have an official status. So that is one of our difficulties as well, and also the difficulties faced by others . . . providing a space for their schooling, when there is no space available and no place to study."* EP02

This participant explained that his learning centre uses a participatory approach for capacity building via peer-instructors, creating a group of trained adolescents to instruct younger children on basic lessons. Due to resource constraints, learning centres are unable to accommodate all children, permitting only the eldest child to attend with the hope that they would instruct their younger siblings. Nevertheless, the NGO activists emphasise that this community's educational needs remain largely unmet.

*Undocumented indigenous people in the interior.* Even though poorly documented indigenous children have access to government schools through the Zero Reject Policy, dropout rates for the indigenous are higher than the national average. The MOE has designed a contextualised curriculum for these communities. In addition, a special model school (K9) offers residential facilities for indigenous children to minimise dropouts [3]. Interviewees informed that even with these provisions, geographical and cultural barriers, and lower school readiness might hinder access to quality education for indigenous children.

## Discussion

The lack of legal identity and non-recognition by the State is the root cause of vulnerability, experienced uniformly by undocumented populations in Malaysia. Without identity documents, an undocumented person is not eligible to enjoy a broad range of human rights including freedom of movement and access to health, education and social services provided by the State, commonly seen as citizenship entitlements [83,84].

Yet, in line with the intersectionality theory [26], it is important to recognise the diversity within the non-citizen populations—they are not homogenous. Certain groups experience more privileged status, while others are more disadvantaged. For example, the Cham refugees escaping the Khmer Rouge in Cambodia, and Bosnian refugees were granted permanent resident status in Malaysia and allowed to integrate locally [55,85–87]. Inconsistencies in treatment may relate to the political context, the community's cultural and religious ties with Malaysia or their socio-economic status in their home country [88]. In this paper, we highlight the disparities experienced by Rohingya refugees. Despite being the largest refugee group in Malaysia, the Rohingya have experienced a protracted refugee situation [89,90], being denied residency, work, and education rights, while being continually marginalised in the Malaysian context. Similarly, Filipino migrants in Sabah have a complex migration history initially recognised as refugees but are now receiving little protection [91,92]. Their status is contrasted with that of Indonesian migrant workers that are supported by employers and the Indonesian Consulate in Sabah [93].

For undocumented children with Malaysian parentage, the lack of legal identity can be partially attributed to gender-discriminatory citizenship laws. Malaysia is one of three countries besides the Bahamas and Barbados, that deny men equal rights in conferring citizenship to

children born outside of a legitimate marriage. Also, Malaysia is one of the twenty-five countries to deny women the right to pass their citizenship to their children on an equal basis with men [94]. The Federal constitution does not allow automatic citizenship for children born abroad to Malaysian women married to foreign spouses, despite Malaysian men being able to do so. While Malaysian parents can apply for citizenship for their children, decisions may take years and rejections are common. Gender discriminatory citizenship laws place children at risk of statelessness if they are unable to obtain citizenship from the other parent [95,96]. Nevertheless, children with citizenship claims have more options compared to other undocumented children. The 'Zero Reject Policy' was launched to ease public school entry for undocumented children of Malaysian parentage, giving parents two years to produce Malaysian identity documents for their children [73]. Unfortunately, uniform implementation of this policy has not been ensured on a national basis [97,98], with some schools refusing admittance of stateless children without documents to prove Malaysian citizenship.

Children born in Malaysia to non-citizen parents or those with undetermined parentage receive few entitlements [99,100]. The future is particularly bleak for undocumented and stateless children without identity documents or pathways to citizenship, as they have little prospect for post-secondary education or formal employment. The policy environment may be influenced by the lack of public sympathy and political will towards improving the rights of undocumented children without Malaysian parentage. The sizable population of undocumented persons in Sabah, despite some being born and raised locally, garners little positive attention from locals as they are perceived as outsiders competing for limited jobs and are blamed for a range of social ills [52,101]. Also, the state government regularly shuts down community learning centres [102–104].

For the refugee communities in Peninsular Malaysia, the alternate education system appears to be more structured with partial oversight from the UNHCR. Several centres prepare children to take international school-leaving examinations towards future resettlement. However, since learning centres aren't regulated by the government, there are inconsistencies in terms of syllabus, teachers, and facilities. Certain communities may be better able to navigate the alternate education system, while marginalised communities like the Rohingya have unequal access. Refugee children have even entered private universities on scholarships [105,106]. However, university enrolment is limited by administrative requirements of passports and student passes which many refugees have difficulty obtaining, recognition of prior learning and prohibitive costs [107,108].

Situating refugee education in its global context, global trends towards protracted situations affecting 15.9 million refugees or 78% of all refugees as of 2018 [109] forces us to rethink the suitability of short-term education responses that are neither certified nor monitored by the host countries. As such, the UNHCR Refugee Education 2030 strategy [110] envisions the inclusion of refugees and stateless children into the national education system as the best option for persons of concern and their hosting communities. The strategy discourages investment in informal education as substitutes for formal education without accredited learning while acknowledging the importance of collaboration with Ministers of Education and the financial and technical support required from the international community [110].

Providing education to all children from a young age is a positive move that would allow children to thrive and eventually contribute to the nation [111,112]. It is estimated that if education is provided to refugees at par with the Malaysian population, refugee contributions to the GDP would increase to over RM6.5 billion annually by 2040, with annual tax contributions of over RM250 million [113].

Successful examples of nations that integrate non-citizen children into their national education systems may be found among European Union member states and countries

that are party to the United Nations CRC, in addition to having national laws that ensure protection and schooling for all children. Comparative studies in these settings have shown that the early inclusion of children in regular classes provides the best chances for school success. Whereas educating refugee children in a parallel education system often has poorer results, with children dropping out or not attending school at all [114,115]. Another example to emulate would be Thailand's progressive "Education for All" (EFA) policy which mandates 15 years of free education for all children, whether they are Thai nationals, migrants, or stateless children. The Thai government sponsors school fees and provides subsidies for books, school supplies, extra-curricular activities, and uniforms. Yet the uptake is low for complex reasons including migrant families' persistent financial struggles with school-related expenses and the difficulty in addressing the multilingual and diverse learning needs of non-citizen children [116–118]. Thus, considerable planning and investment are necessary to accommodate and integrate non-citizen children into the Malaysian public school system in the long term. Other good practices range from mainstreaming host country curriculum in alternative education or accrediting informal education with national or regional equivalencies, and regional initiatives such as the Djibouti Declaration and Plan of Action on Refugee Education comprising eight countries [119–121].

This study has several strengths. We used a framework analysis methodology, which is flexible yet systematic, enabling us to summarise and chart data, while keeping the context of each case, thereby allowing for thick descriptions [32,122,123]. This study is a unique examination of undocumented, stateless, and non-citizen children in Malaysia, exploring their identity and access to education. The children discussed here belong to hidden populations that are poorly accounted for in national statistics. We hope that this paper contributes to a better understanding that allows for the identification of intervention points for future action.

This study has several limitations. We were unable to conduct fieldwork to observe the environment and surroundings of learning centres due to the strict lockdowns enforced during the COVID-19 pandemic. Interviews were mostly conducted virtually of stakeholders representing communities studied. We had difficulties interviewing non-citizens, particularly from East Malaysia, as travel restrictions limited study participants to those with the means to engage in online interviews. Nevertheless, we were able to triangulate our findings by conducting a document review and interviewing diverse key informants with experience in marginalized undocumented communities in Malaysia. Also, the qualitative nature of this study does not allow the generalisation of findings beyond this setting.

## Conclusion

The alternative education system in Malaysia is highly responsive and serves the diverse educational needs of non-citizen communities in Malaysia. Yet, many children are still left behind. Implementing a rights-based approach towards education would mean allowing all children equal opportunity to access and thrive in high-quality schools. Education is essential for the social and emotional well-being of children and would yield a significant return on investment as adults enter the workforce at higher wages and with greater skills.

We would caution against the stringent regulation of informal education, as this may result in learning centres being shut down. The government should instead support learning centres in providing quality education by allowing the use of the national syllabus for pre-school, primary and secondary education, and allowing for all children to sit for Malaysian school-leaving examinations. This would pave the way for post-secondary and tertiary education in Malaysia or elsewhere, and better job prospects.

## Supporting information

**S1 File. Interview guide.**
(PDF)

## Author Contributions

**Conceptualization:** Tharani Loganathan, Zhie X. Chan.

**Data curation:** Tharani Loganathan, Zhie X. Chan, Fikri Hassan.

**Formal analysis:** Tharani Loganathan, Zhie X. Chan, Fikri Hassan, Zhen Ling Ong.

**Funding acquisition:** Tharani Loganathan.

**Investigation:** Tharani Loganathan, Hazreen Abdul Majid.

**Methodology:** Tharani Loganathan.

**Project administration:** Fikri Hassan.

**Software:** Tharani Loganathan, Zhie X. Chan.

**Supervision:** Hazreen Abdul Majid.

**Validation:** Zhie X. Chan, Fikri Hassan, Zhen Ling Ong.

**Visualization:** Zhie X. Chan, Zhen Ling Ong.

**Writing – original draft:** Tharani Loganathan, Fikri Hassan, Zhen Ling Ong.

**Writing – review & editing:** Tharani Loganathan, Zhen Ling Ong, Hazreen Abdul Majid.

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
