## [Decision Letter · Decision Letter 0]

13 Dec 2021

PONE-D-21-36078Undocumented: An examination of legal identity and education provision for children in MalaysiaPLOS ONE

Dear Dr. Tharani Loganathan,

Thank you for submitting your manuscript to PLOS ONE. After careful consideration, we feel that it has merit but does not fully meet PLOS ONE’s publication criteria as it currently stands. Therefore, we invite you to submit a revised version of the manuscript that addresses the points raised during the review process.

We look forward to receiving your revised manuscript.

Kind regards,

Rogis Baker, Ph.D

Academic Editor

PLOS ONE

Journal Requirements:

2. We note that Figure 1 in your submission contain [map/satellite] images which may be copyrighted. All PLOS content is published under the Creative Commons Attribution License (CC BY 4.0), which means that the manuscript, images, and Supporting Information files will be freely available online, and any third party is permitted to access, download, copy, distribute, and use these materials in any way, even commercially, with proper attribution. For these reasons, we cannot publish previously copyrighted maps or satellite images created using proprietary data, such as Google software (Google Maps, Street View, and Earth). For more information, see our copyright guidelines: http://journals.plos.org/plosone/s/licenses-and-copyright.

a) You may seek permission from the original copyright holder of Figure 1 to publish the content specifically under the CC BY 4.0 license.  

Reviewers' comments:

Reviewer's Responses to Questions

**Comments to the Author**

1. Is the manuscript technically sound, and do the data support the conclusions?

Reviewer #1: Yes

Reviewer #2: Yes

2. Has the statistical analysis been performed appropriately and rigorously? 

Reviewer #1: N/A

Reviewer #2: N/A

3. Have the authors made all data underlying the findings in their manuscript fully available?

Reviewer #1: Yes

Reviewer #2: Yes

4. Is the manuscript presented in an intelligible fashion and written in standard English?

Reviewer #1: Yes

Reviewer #2: Yes

5. Review Comments to the Author

Reviewer #1: Thank you for the opportunity to review this manuscript. In general, this paper highlights one of the important issues in education at the moment, which accentuates its relevance for publication. However, in order to enhance the scholarly rigour of the manuscript, the authors could consider refining their methodology part:

1. With regards to the data collection procedures, it would be more structured if the authors could explain how they developed the interview protocol and how interview questions were developed in alignment with the research objective.

2. Readers might want to focus more on "framework analysis" as the main method for qualitative data analysis. It is suggested that the authors include the relevant references in the discussion.

3. The authors need to explain how the data were triangulated and how their qualitative data had been validated.

Reviewer #2: This study uses a qualitative design; hence, the appropriateness and rigour of statistical analysis are inapplicable. However, the qualitative method and the results upon which they are based are sound and appropriate.

Although the authors state that the data are available without restrictions, no link or location is provided to access the data.

This manuscript presents data about undocumented children and their educational opportunities, or lack thereof, in Malaysia. It details the many categories of undocumented children and the limited or perhaps restricted educational settings available to them. It focuses the light on the plight of these children and their resulting ill-effect on their future. As the authors allude to, educational opportunities for undocumented children are governed by the laws of each country and range from accommodating to far less accommodating. While this manuscript presents the plight of these children in Malaysia and will likely be beneficial to readers in Malaysia, it may provide readers from other countries with a list of what not to do.

The manuscript is well-written, detailed, informative, and easy to read. Overall, the methodology, the results, and the conclusions are sound. The only issue is the limited number of participants. For such countrywide conclusions, relying on single-digit respondents in a primary role and very few non-citizens make the results and conclusions, while still useful, rather limited. The limitations imposed by these low numbers and the reasons for these low numbers are not addressed in the manuscript. I shall leave the decision whether to request a minor revision to the editor.

6. PLOS authors have the option to publish the peer review history of their article (what does this mean?). If published, this will include your full peer review and any attached files.

Reviewer #1: No

Reviewer #2: **Yes: **Dr. Ali M. AL-Asadi

---

## [Author Response · Author response to Decision Letter 0]

8 Jan 2022

Response to reviewers

Journal Requirements

 We have reviewed the manuscript and it meets PLOS ONE’s style requirement, including file naming.

1) Figure 1 was replaced with a new figure and permission was granted from the original copyright holder to publish and edit the content specifically under the CC BY 4.0 license. The completed content permission form has been uploaded as an ‘Other file’ with our submission. 

The copyrighted figure has been acknowledged as: Reprinted from http://www.ofo.my/ under a CC by license, with permission from OFO Tech Sdn Bhd, original copyright 2021.

We have also inserted a note to Figure 1, to ease understanding of the international readers.

The figure caption is as below:

Fig 1. Different categories of undocumented and non-citizens children in Malaysia by location and legal identities

Note: Malaysia is comprised of Peninsular Malaysia and East Malaysia, separated by the South China Sea. Sabah is one of the states in East Malaysia.

Reprinted from http://www.ofo.my/ under a CC by license, with permission from OFO Tech Sdn Bhd, original copyright 2021.

2) Captions for Supporting Information Files at the end of our manuscript has been inserted and in-text citations have been matched accordingly.

3) The reference list has been reviewed and it is complete and correct, we have not cited any papers which have been retracted. No changes have been made to the reference list.

 

Reviewer #1:

1. With regards to the data collection procedures, it would be more structured if the authors could explain how they developed the interview protocol and how interview questions were developed in alignment with the research objective.

Thank you for the suggestions. We have included the following in the Materials and Methods section (Page 6, Line 110 -116):

Semi-structured interview guides were developed based on our desk review. The interview guides contained introductory questions to understand and contextualise non-citizen groups and open questions on education policies relevant to non-citizen children. These guides were developed for 3 main categories of interviewees:(a) teachers and educators, (b) parents and migrant representatives and (c) policymakers and high-level stakeholders. We customised interviews according to the background of the interviewee. Minor improvements were made after our initial reflections from the earlier interviews. See S1 File for interview guides.

2. Readers might want to focus more on "framework analysis" as the main method for qualitative data analysis. It is suggested that the authors include the relevant references in the discussion.

We describe framework analysis conducted in the Materials and Methods section:

Study Design subsection (Page 6, Line 98-100):

Framework analysis was conducted to identify, define, and contextualise different categories of undocumented children at risk of education exclusion in Malaysia.

Data collection and analysis subsection (Page 8, Line 155-163)

We conducted framework analysis; a qualitative methodology suited to applied policy research. Findings from the desk review and in-depth interviews were analysed using five steps: familiarisation, identifying a thematic framework, indexing, charting and interpretation [1]. This descriptive analysis allowed us to categorise and contextualise undocumented children at risk of education deprivation by location (Overall Malaysia, Peninsular Malaysia and Sabah, East Malaysia) and legal identities, define the concept of being ‘undocumented’, map the types of education provision and provide an analysis of key issues and policies on education provision that are specific or overlapping for each group of children.

We have included the following in the Discussion section (Page 41, Line 859-862):

We used a framework analysis methodology, which is flexible yet systematic, enabling us to summarise and chart data, while keeping the context of each case, thereby allowing for thick descriptions[1-3]. 

3. The authors need to explain how the data were triangulated and how their qualitative data had been validated.

We have included the following in the Materials and Methods section

Page 8, Line 146 -148:

Most interviews were conducted by at least 2 researchers, with one researcher leading and the other observing and taking field notes. 

Page 9, Line 163- 166:

Interviews with stakeholders from different backgrounds and concurrent desk review allowed for the triangulation of findings. Qualitative data were validated through regular discussion with the entire team, as well as member checks, audit trails and giving attention to minor themes.

Reviewer #2: 

1. Although the authors state that the data are available without restrictions, no link or location is provided to access the data.

We have changed the data availability statement on the PLoS submission system to ‘Some restrictions will apply’. To protect the respondent's anonymity, ethical constraints prevent the data set from being made public. Data may contain personally identifiable or sensitive respondent information. Participants in the study are vulnerable populations whose data when combined, could become identifying due to indirect identifiers (such as ethnicity, location, etc.). 

Data requests can be made from the University of Malaya Research Ethics Committee (UMREC, reference number: UM. TNC2/UMREC- 848) for researchers who meet the criteria for access to confidential data. 

All information collected for this study will be kept safely for a minimum period of 5 years, according to the period prescribed by the Universities’ Ethics Committee. Once the recommended period has lapsed without the need for any further analysis and audits, all electronic data will be deleted.

2. The manuscript is well-written, detailed, informative, and easy to read. Overall, the methodology, the results, and the conclusions are sound. The only issue is the limited number of participants. For such countrywide conclusions, relying on single-digit respondents in a primary role and very few non-citizens make the results and conclusions, while still useful, rather limited. The limitations imposed by these low numbers and the reasons for these low numbers are not addressed in the manuscript. I shall leave the decision whether to request a minor revision to the editor.

As we used qualitative methodology for in-depth interviews, we were less concerned about the sample size, but in assuring that sufficient data was collected to meaningfully answer our research questions and achieve thematic saturation. Thus, we do not consider it a limitation that our sample size was ‘small’.

However, we acknowledge that the lack of non-citizen participation is a study limitation. 

We have included the following in the Discussion section (Page 41, Line 870-874):

We had difficulties interviewing non-citizens, particularly from East Malaysia, as travel restrictions limited study participants to those with the means to engage in online interviews. Nevertheless, we were able to triangulate our findings by conducting a document review and interviewing diverse key informants with knowledge on marginalized undocumented communities in Malaysia. 

 

References

1. Srivastava, Aashish, Thomson, Stanley. Framework analysis: a qualitative methodology for applied policy research. 4 Journal of Adminstration and Governance. 2009;72.

2. Popay J, Rogers A, Williams G. Rationale and standards for the systematic review of qualitative literature in health services research. Qual Health Res. 1998;8(3):341-51. Epub 1999/11/11. doi: 10.1177/104973239800800305. PubMed PMID: 10558335.

3. Gale NK, Heath G, Cameron E, Rashid S, Redwood S. Using the framework method for the analysis of qualitative data in multi-disciplinary health research. BMC Med Res Methodol. 2013;13:117. Epub 2013/09/21. doi: 10.1186/1471-2288-13-117. PubMed PMID: 24047204; PubMed Central PMCID: PMCPMC3848812.

---

## [Editor Report · Decision Letter 1]

19 Jan 2022

Undocumented: An examination of legal identity and education provision for children in Malaysia

PONE-D-21-36078R1

Dear Dr. Tharani Loganathan,

We’re pleased to inform you that your manuscript has been judged scientifically suitable for publication and will be formally accepted for publication once it meets all outstanding technical requirements.

Kind regards,

Rogis Baker, Ph.D

Academic Editor

PLOS ONE
---

## [Editor Report · Acceptance letter]

24 Jan 2022

PONE-D-21-36078R1 

Undocumented: An examination of legal identity and education provision for children in Malaysia 

Dear Dr. Loganathan:

I'm pleased to inform you that your manuscript has been deemed suitable for publication in PLOS ONE. Congratulations! Your manuscript is now with our production department. 

Kind regards, 

on behalf of

Dr. Rogis Baker 

Academic Editor

PLOS ONE